# Development of a Single-Neurosphere Culture to Assess Radiation Toxicity and Pre-Clinical Cancer Combination Therapy Safety

**DOI:** 10.3390/cancers15204916

**Published:** 2023-10-10

**Authors:** Bedika Pathak, Taylor E. Lange, Kristin Lampe, Ella Hollander, Marina Oria, Kendall P. Murphy, Nathan Salomonis, Mathieu Sertorio, Marc Oria

**Affiliations:** 1Center for Fetal and Placental Research, Cincinnati Children’s Hospital Medical Center (CCHMC), Cincinnati, OH 45229, USA; bedika.pathak@cchmc.org (B.P.); kristin.lampe@cchmc.org (K.L.);; 2Department of Cancer Biology, University of Cincinnati College of Medicine, Cincinnati, OH 45267, USA; taylor.miller2@cchmc.org; 3Division of Oncology, Cancer and Blood Diseases Institute, Cincinnati Children’s Hospital Medical Center (CCHMC), Cincinnati, OH 45229, USA; 4Department of Orthopedic Surgery, University of Cincinnati College of Medicine, Cincinnati, OH 45267, USA; 5Division of Biomedical Informatics, Cincinnati Children’s Hospital Medical Center (CCHMC), Cincinnati, OH 45229, USA; nathan.salominis@cchmc.org; 6Departments of Pediatrics and Bioinformatics, University of Cincinnati, Cincinnati, OH 45221, USA; 7University of Cincinnati Cancer Center, Cincinnati, OH 45267, USA; sertormu@ucmail.uc.edu; 8Department of Radiation Oncology, University of Cincinnati College of Medicine, Cincinnati, OH 45267, USA; 9University of Cincinnati Brain Tumor Center, Cincinnati, OH 45219, USA

**Keywords:** neural progenitor cells, radiation, radiation toxicity, astrogenesis, neurogenesis

## Abstract

**Simple Summary:**

Exposure to radiation during nuclear catastrophes, natural sources such as space travel, or during cancer treatment can lead to central nervous system toxicity. The high sensitivity of neural progenitor cells (NPCs) to radiation poses a significant obstacle to dose escalation for the treatment of brain cancer such as glioblastoma, yet studies suggest that higher (dose-escalated) radiation therapy (RT) could enhance local control. Unfortunately, this is presently infeasible due to the toxicity to neighboring normal tissue; thus, there is a critical need to understand the cellular mechanisms that determine the extent of radiation-induced toxicity in normal NPCs, to develop CNS toxicity radiation counter measures or develop new safe brain cancer chemo–radiation therapy strategies.

**Abstract:**

Radiation therapy (RT) is a crucial treatment modality for central nervous system (CNS) tumors but toxicity to healthy CNS tissues remains a challenge. Additionally, environmental exposure to radiation during nuclear catastrophes or space travel presents a risk of CNS toxicity. However, the underlying mechanisms of radiation-induced CNS toxicity are not fully understood. Neural progenitor cells (NPCs) are highly radiosensitive, resulting in decreased neurogenesis in the hippocampus. This study aimed to characterize a novel platform utilizing rat NPCs cultured as 3D neurospheres (NSps) to screen the safety and efficacy of experimental drugs with and without radiation exposure. The effect of radiation on NSp growth and differentiation was assessed by measuring sphere volume and the expression of neuronal differentiation markers Nestin and GFAP and proliferation marker Ki67. Radiation exposure inhibited NSp growth, decreased proliferation, and increased GFAP expression, indicating astrocytic differentiation. RNA sequencing analysis supported these findings, showing upregulation of Notch, BMP2/4, S100b, and GFAP gene expression during astrogenesis. By recapitulating radiation-induced toxicity and astrocytic differentiation, this single-NSp culture system provides a high-throughput preclinical model for assessing the effects of various radiation modalities and evaluates the safety and efficacy of potential therapeutic interventions in combination with radiation.

## 1. Introduction

Radiation therapy (RT) is essential for palliative and curative treatments of central nervous system (CNS) tumors and CNS metastases for carcinomas or leukemias; however, radiation toxicity to the CNS remains a limitation to therapeutic success and long-term patient outcomes [1,2]. Radiation-induced CNS toxicity is a complex phenomenon that is not completely understood. In addition to radiotherapy, exposure to radiation during nuclear catastrophe or natural sources such as long space travel could represent a risk for CNS toxicity [3]. Inflammation and edema due to the activation of CNS resident immune cells, pro-inflammatory signals from senescent cells, and direct damage to the neural compartment all participate in CNS toxicity [4,5].

The two germinal zones in the brain, the hippocampus and subventricular zone (SVZ), contain multipotent neural progenitor cells (NPCs) that generate three major cell lineages: neurons, oligodendrocytes, and astrocytes. NPCs are responsible for normal neural development and as a reserve pool for tissue regeneration [6]. Additionally, NPCs have been reported to be highly radiosensitive, as radiation exposure causes decreased neurogenesis after CNS irradiation, which correlates to cognitive deficits including learning disabilities, abnormal growth, and motor retardation [7]. Inflammatory changes in the microenvironment in the NPC niche cause proliferation inhibition and lead to premature differentiation into astrocytes [8,9,10,11].

Cell cultures derived from neural tissues, including cancer tissues, are essential tools used for understanding the cellular and molecular mechanisms responsible for neural development and function [12]. Furthermore, they can be utilized for screening potential neuro-protective or antitumorigenic compounds to identify novel treatments. High-throughput drug screens are traditionally conducted using monolayer cell culture models, which fail to fully simulate authentic cell interactions; however, cells can also be grown in three-dimensional structures, such as spheroids. Three-dimensional (3D) culture models better reproduce cell and matrix interfaces but are limited by size variability [13]. The development of optimal 3D culture methods for neural and cancer stem cells is critical for demonstrating their applicability in identifying new treatment options and characterizing responses to radiation.

Cognitive and developmental alterations in the brain are primarily induced by radiotoxicity to NPCs during radiation exposure. Despite the sensitivity and toxicity studies on the NPC germinal zones after radiation in rodents, reliable and cost-effective methods to assess different compounds’ effects are still not well developed. This study characterizes the dose effect of radiation exposure to rat NPCs and the potential use of a novel single-neurosphere (NSp) culture method that could be used as a platform for NPC toxicity studies and preclinical drug screenings to evaluate the safety of new drug/radiotherapy treatments for CNS cancers and to test the efficacy of CNS radiation toxicity pharmacological counter measures.

## 2. Materials and Methods

The experimental protocols agreed with the National Institutes of Health Guidelines for Care and Use of Laboratory Animals and were approved by the Institutional Animal Care and Use Committee at Cincinnati Children’s Hospital Medical Center (IACUC 2019-0081).

### 2.1. Tissue Processing and Neurosphere Cultures

The study was performed using timed-pregnant Sprague Dawley rats weighing 200–250 g (Charles River Laboratories, Inc, Wilmington, MA, USA), housed at 22 °C in a standard dark:light cycle (10:14 h) (light 7:00–19:00) with access to water and standard food ad libitum. The mating date was defined as E-1 and plug day as E-0.

Neural progenitor cells (NPCs) were isolated from the brain of E15 Sprague Dawley rat fetuses and cultured in NeuroCult (StemCell Technologies, Vancouver, BC, Canada) basal medium supplemented with 1% Pen/Strep, 1X N-2 supplement, 1X B-27 supplement, 20 ng/mL human epidermal growth factor, 20ng/mL human fibroblast growth factor, and 1:500 dilution of 0.2% heparin solution in ultra-low attachment microplates where they spontaneously formed spheroids (neurospheres; NSp). Culture media was exchanged every 3–4 days and NSp spheroids were dissociated into NPCs every 7–10 days using 0.05% trypsin-EDTA to maintain the cells in their progenitor state.

For NPC expansion and RNA extraction, NPCs were seeded at a density of 1 × 10^5^ cells/well (Corning, New York, NY, USA) 6-well ultra-low attachment microplates or 4 × 10^4^ cells/well in Corning^®^ 24-well ultra-low attachment microplates.

To define culture conditions for individual NSp cultures, NPCs were seeded at various cell densities (100, 200, 400, and 800 cells/well) in Corning^®^ 96-well ultra-low attachment U-bottom microplates and individual NSp growth was measured at 3, 6, and 7 days after culture.

NSp radiation toxicity treatments were administered as pretreatment on day 2 post-seeding (before radiation exposure) and on day 3 after radiation exposure added pre- and post-radiation to the media. NSps were kept in culture for up to 7 days and then analyzed. The treatments tested were ascorbic acid (ASCOR^®^ McGuff Pharmaceuticals, Santa Ana, CA, USA #67157) at a dose of 200 µM and Recombinant Noggin (Prepotech, Thermo Fisher Scientific, Waltham, MA, USA #120-10C) at 5 µM.

### 2.2. Radiation Exposure

The effect of radiation on NSp growth and differentiation was measured after 0, 0.5, 1, or 2 Gy of photon radiation (day = 3 after NPCs seeding) administered with a CellRad+ benchtop X-ray irradiator (Precision, Madison, CT, USA). Plates containing NSps were placed on the rotating plate holder in the irradiation chamber using the radiation field template of the holder. The plates were irradiated at the indicated dose at an average dose rate of 1.2 Gy/min (150 KeV, 6 mA). Dose rate and accurate dose delivery were checked before and during radiation using the calibrated integrated dosimetry system. Culture media was changed after radiation.

### 2.3. Apoptosis Quantification Using Flow Cytometry

Dissociated cells from neurospheres previously grown and treated in 6 well plates were stained with Annexin V-APC and propidium iodide (eBioscience, Thermo Fisher Scientific, Waltham, MA, USA) following the manufacturer’s protocol, utilizing Annexin V staining buffer. Neurospheres from one well of six well plates were treated as one replicate for each condition tested. A total of 5 to 6 independent replicates were performed per condition tested. Samples stained with Annexin V-APC + PI, as well as unstained and single-stained controls, were analyzed using a BD Canto Flow cytometer (BD Biosciences, Franklin Lakes, NJ, USA). Compensation parameters were established using the BD DIVA acquisition software prior to the acquisition of stained samples. The proportion of apoptotic cell death (Annexin V+ and PI+ or PI− cells) and non-apoptotic cell death (PI+ and AnnexinV-cells) was quantified from all singlet cells using FlowJo (v10.9), after fluorescence compensation and gating based on single-color stained samples.

### 2.4. Immunostaining

On day 4 post-radiation, the neurospheres were collected and cytospun onto slides. The slides were fixed with 4% paraformaldehyde for 30′ at room temperature. NSps were permeabilized with 0.5% Triton X-100 (Sigma Aldrich, St. Louis, MO, USA) in phosphate-buffered saline (PBS) for 15 min at room temperature before non-specific binding was blocked for 1 h with 5% BSA in PBS at room temperature. Slides were then incubated overnight at 4 °C in a humidity chamber with the following primary antibodies: anti-Ki67 (Abcam, Waltham, MA, USA) (1:500), anti-GFAP (Abcam, Waltham, MA, USA) (1:500), and anti-Nestin (BD Biosciences, Franklin Lakes, NJ, USA) (1:50). The slides were washed and incubated for 1 h with Alexa Fluor 488-, Alexa Fluor 568-, or Alexa Fluor 647-conjugated secondary antibodies (Life Technologies, Eugene, OR, USA) (1:1000) at room temperature, protected from light in a humidity chamber. Finally, the slides were washed, covered with mounting media containing DAPI (Southern Biotech, Birmingham, AL, USA), and visualized with a Nikon fluorescent microscope (Nikon Inc., Melville, NY, USA).

### 2.5. Immunolabeled Cells and Area Quantification

GFAP+, Nestin2+ immuno-stained area measurements, and Ki67+ cell counts were performed using NIS Elements AR 4.5 software (Nikon Instruments Inc). Quantification was conducted using more than eight random high-magnification images per neurosphere. This analysis was conducted in 4–6 neurospheres per group and the experiments were repeated three times. Cell count data are reported as the percentage of immuno-positive cells compared to the total number of cells in each area. Area is reported as the positively stained area in square pixels as a percentage of the total area. All quantifications were performed by an investigator blinded to the experimental groups.

### 2.6. RNA Extraction and RT-qPCR Analysis

RNA was extracted from NSps using TRIzol and pipet homogenization. Glycogen was added to the TRIzol sample after homogenization to aid in RNA recovery. RNA quantification was assessed using Epoch Biotek spectrophotometer (Biotek Instruments, Winooski, VT, USA). Utilizing the RT2 First Strand Kit (QIAGEN, Germantown, MD, USA), a 0.5 µg RNA sample was reverse-transcribed into cDNA. A 25 ng cDNA sample was then used as a template for RT-qPCR, employing TaqManR gene expression assays (Applied Biosystems, Foster City, CA, USA) (Appendix A) in the Applied Biosystems StepOnePlus Real-Time PCR System. Samples were run in duplicate for target genes and were normalized using HPRT1 as an endogenous control. Relative quantification of transcript expression was performed using the 2^−ΔΔCt^ method, where Ct represents the threshold cycle.

### 2.7. Neurospheres’ Volume Quantification

Brightfield microscopy images were taken at 0, 2, and 4 days after radiation and images were analyzed using ImageJ I(J1.46r) to determine sphere volume. Quantification was conducted using 16–20 neurospheres per group. Sphere area was determined using ImageJ. Then, radius was calculated as (sphere area/(4 × 3.14))^(0.5). After we had the radius, we calculated volume as: (4/3)(3.14)(radius)^3. Cell volume data and all quantifications were performed by an investigator blinded to the experimental groups.

### 2.8. cDNA Library Preparation and Sequencing

Directional polyA bulk RNA-sequencing was performed using established protocols as previously mentioned (PMID: 31120332 and 31420676) by the Genomics, Epigenomics, and Sequencing Core at the University of Cincinnati. To summarize, the quality of the total RNA was QC analyzed using a Bioanalyzer (Agilent, Santa Clara, CA, USA). After passing QC, 1 µg of total RNA was used in the NEBNext Poly(A) mRNA Magnetic Isolation Module (New England BioLabs, Ipswich, MA, USA) to isolate polyA RNA for library preparation. The polyA RNA was further enriched using SMARTer Apollo automated NGS library prep system (Takara Bio USA, Mountain View, CA, USA). Next, NEBNext Ultra II Directional RNA Library Prep kit (New England BioLabs, Ipswich, MA, USA) was used for library preparation under PCR cycle number 8 and the resulting library QC’d and quantified via Qubit quantification (ThermoFisher, Waltham, MA, USA). Using single read 1 × 85 bp settings on a NextSeq 2000 sequencer (Illumina, San Diego, CA, USA), individually indexed libraries were proportionally pooled and sequenced. The individual samples were given unique identifiers during processing by the Genomics, Epigenomics, and Sequencing Core and these identifiers were kept consistent throughout the manuscript. The identifiers follow the nomenclature “MO#” indicating the initials of the corresponding author and the number of the sample.

### 2.9. RNA-Seq Data Processing and Analysis

Paired-end reads were aligned to rat genome build Rn7 with STAR version 2.6.1. Principal component analysis (PCA) highlighted potential batch effects. One outlier sample was identified from the control treatment group (MO2) and was removed. Kallisto was used to generate TPM, which were then processed through NOISeq version 2.42.0 for batch correction. AltAnalyze version 2.1.4.4 (EnsMart106 database) was then used for differential expression analysis on the batch-corrected data matrix. The AltAnalyze supplied empirical Bayes moderated *t*-test was performed followed by Benjamini–Hochberg adjustment for false discovery (FDR). Final threshold cutoffs were an adjusted *p*-value ≤ 0.05 (FDR) and fold change ≥ 1.5. Pathway enrichment was also performed in AltAnalyze using the integrated GO-Elite pathway analysis tool and additionally with the online tool g:Profiler (https://biit.cs.ut.ee/gprofiler/gost accessed on 13 December 2022). Venn diagrams were created in AltAnalyze using all differentially expressed genes with an adjusted *p*-value ≤ 0.05 (FDR) and fold change ≥ 1.5. The volcano plots were generated with all expressed genes using the Enhanced Volcano package in R version 4.2.1 (https://github.com/kevinblighe/EnhancedVolcano). Significantly upregulated genes ≥ a linear 1.5-fold increase were indicated in red and downregulated in blue. The protein–protein interaction (PPI) networks were created using STRING version 12.0 [14] and a protein–protein interaction score of at least 0.4 (https://string-db.org/). Predicted PPIs were considered significant if the PPI enrichment *p*-value was ≤ 0.05.

RNAseq data were deposited in the Sequence Read Archive (SRA) and can be found via GEO accession number (GSE242899).

### 2.10. Statistical Analysis

Unless otherwise specified, all statistical analysis and graphs were performed in Graph Pad Prism 9 software (GraphPad Software Inc., La Jolla, CA, USA). Differences among multiple groups were analyzed using one-way analysis of variances (ANOVA) using Tukey’s post hoc test. Differences among the same group at different time points were analyzed using two-way analysis of variance (ANOVA) using Tukey’s post hoc test. Results are reported as means ± standard error (SE) for the relative gene expression (2^−ΔΔCt^) and means ± standard deviation (SD) for the cell counting analysis. A *p*-value ≤ 0.05 was considered statistically significant.

## 3. Results

### 3.1. NSp Cultures Recapitulate NPC-Defining Characteristics

This study aimed to create a reliable single-neurosphere (NSp) culture system to evaluate the responses of NPCs to radiation and to test the safety or benefit of compounds and/or potential therapies. To determine the optimal NPC cellular seeding density required us to form a single NSp by 3 days post-seeding, we seeded 100, 200, 400, and 800 rat NPC cells/well. We observed a linear increase in the volume of the NSps over time in all seeding densities tested; however, 400 and 800 cells/well gave the largest NSps, nearly doubling in initial size by the day 7 endpoint (Figure 1A). The optimal NSp size was determined by seeding 200 cells/well at day 7 post-seeding; the 400 and 800 cells/well NSps were too large and dense, potentially not allowing proper nutrients to reach the center of the NSp (Figure 1B).

Under the specified culture conditions, NSps and the NPCs were found to express stemness markers such as Nestin (green) and Musashi (red). Immunostaining showed a high expression of Ki67 (red), a marker of cell proliferation that is expressed in NPCs (Figure 1C). Taken together, we demonstrated that single-NSp cultures exhibit the same defining characteristics of multiple NPCs in culture.

### 3.2. Radiation Induces Apoptosis after Radiation Exposure

As radiation exposure is known to cause radiotoxicity in traditional NPC culture, we assessed the effect of radiation treatment on apoptosis, necrosis, and cell proliferation in the NSps. We stained NSps 4 days after radiation and used flow cytometry to measure apoptosis and necrosis. We detected a significant increase in apoptosis in the 2 Gy irradiated NSps as compared to non-irradiated samples (Figure 2A). There were no significant differences in necrosis between any of the tested conditions (Figure 2B).

### 3.3. Radiation Treatment Increases NPC Differentiation in Neurospheres

To understand the possible mechanism by which radiation exposure in rat NSps causes radiotoxicity, changes in global gene expression were measured using bulk RNA-sequencing comparing non-irradiated controls (C) with NSps exposed to 1 Gy radiation alone (CGy). Principal component analysis (PCA) demonstrated the separation of the irradiated samples from the controls indicating that radiation treatment significantly altered the transcriptomic profile of NSps and accounted for >90% of the variation (Figure 3A,C). A gene expression threshold was set at ±1.5-fold change with an adjusted *p*-value ≤ 0.05. A total of 396 genes were differentially expressed upon radiation treatment (209 upregulated and 97 downregulated) (Figure 3B,C).

Hierarchical clustering was performed followed using gene ontology (GO) pathway analyses with AltAnalyze’s built-in GO-Elite. By overlaying the biological process pathways with the clustered heatmap, the overall response to radiation was assessed and reinforced that there was a clear separation in transcriptomic signature between the two groups (Figure 3C). Radiation treatment upregulated genes enriched for biological process pathways including adhesion, cell signaling, response to ROS, and proliferation but downregulated genes involved in metabolism and development. Additionally, to independently validate the pathway analysis using a separate tool, the differentially expressed genes were analyzed using g:Profiler. The pathway analysis results from g:Profiler GO: Biological Processes, GO: Cellular Components, GO: Molecular Function, and KEGG pathways supported the GO-Elite findings and indicated that radiation differentially regulates pathways involved in cellular metabolism, development, response to stress, and most strikingly, differentiation in NSps (Appendix A).

To determine if the differentially expressed genes encode proteins that have direct (physical) or indirect (functional) associations, the top 20 upregulated and top 20 downregulated genes were input into STRING (Appendix A). Using a medium interaction confidence score of at least 0.4, the resulting protein–protein interaction (PPI) network consisted of a total of 14 connected genes that separated into two distinct clusters: one consisting of twelve proteins and a smaller cluster consisting of two proteins (Figure 4). The smaller cluster contained Bnip3 and Pdk1. Expression of these genes can be induced during hypoxia and the resulting proteins are involved in controlling mitophagy [15,16]. The larger cluster of 12 genes/proteins primarily play roles in modulating differentiation, particularly of astrocytes and neurons but also cell proliferation, response to hypoxia, and cell death (Appendix A). Collectively, these results provide evidence that radiation induces effects on proliferation and differentiation in NSp cultures in response to radiation-induced damage, such as hypoxia and inflammation, to promote cell survival.

### 3.4. Neurogenesis Adaptation after Radiation of Single NSps

NSCs’ exposure to radiation causes permanent inhibition of NSC proliferation in the neurogenic niche [12] and leads to differentiation of NPCs into glial fate [8]. In previous works by Korirova J et al., radiation exposure induced neuronal differentiation increasing expression of Tubb3 and of the astrocyte markers GFAP [12]. To elucidate the changes in gene expression that could alter the neural progenitor cell commitment after radiation exposure, we checked the expression of genes within the neurogenesis gene ontology (GO:002208), and the three sub-classifications within neurogenesis: oligodendrocyte differentiation (GO:0048709), astrocyte differentiation (GO:0048708), and neuron differentiation (GO:00330182). Out of 174 total genes that make up the oligodendrocyte differentiation pathway, 4 genes were differentially expressed in the CGy NSps as compared to the C NSps, and out of 424 total genes in the astrocyte differentiation pathway, 8 were differentially expressed. Within the neuron differentiation pathway, we identified 56 genes dysregulated between CGy and Control from 4046 total gene annotations (Appendix A).

To validate the effects of radiation on the growth potential and differentiation of NSps, we exposed NSps to 0.5, 1, or 2 Gy X-ray radiation three days after seeding and monitored NSp formation. NSp formation was significantly reduced upon treatment with 0.5 Gy radiation and completely prevented with 1 and 2 Gy treatment (Figure 5A). We also observed decreased cell proliferation and cell stemness, as quantified from Ki67 and Nestin immunofluorescence staining, respectively (Figure 5B,C). Moreover, we detected an increase in GFAP protein expression concomitant with the increasing radiation dose indicating an increase in differentiation of NPCs into astrocytes. In accordance with the protein expression results, there was a radiation dose-dependent upregulation of genes involved in astrocyte differentiation such as GFAP, ALDH1L1, AQP4, BMP4, and BMP2 (*** *p* < 0.001) (Figure 6). A progressive downregulation of Nestin gene expression was detected with the increasing radiation dose becoming significantly different than untreated cells at 2 Gy (*** *p* < 0.001). Taken together, these results indicate that radiation treatment promotes astrocyte differentiation of NPCs in neurosphere cultures, which recapitulates reports from published in vivo studies.

### 3.5. Inhibition of BMP Activity or Supplementation with Antioxidants Reduces Radiotoxicity in Neurospheres

To identify compounds with radioprotective properties, we pretreated NSp cultures with various compounds and measured changes in gene expression with or without radiation treatment. We pretreated NSps with Recombinant Noggin at 5 µM, ascorbic acid at 200 µM, or vehicle and then irradiated with 1 Gy radiation at 3 days post-seeding. As astrocyte differentiation and GFAP expression rely on BMP activity, we pretreated NSps with the BMP signaling inhibitor Noggin [17]. We also tested ascorbic acid due to its antioxidant properties and its known role in the maintenance of NPC pluripotency [18]. At 7 days post-irradiation, RNA was extracted from NSps from the six treatment groups (control (C); control + radiation (CGy); Noggin treatment (N); Noggin + radiation (NGy); ascorbic acid treatment (A); ascorbic acid + radiation (AGy)) and gene expression measured via bulk RNA sequencing. The PCA showed distinct separation between all groups except for the CGy and NGy samples, which clustered together (Figure 7, left panel). This may suggest that treatment with Noggin minimally changes the transcriptome after radiation treatment. Hierarchical clustering analysis indicated that the non-irradiated control (C) and both ascorbic-acid-treated groups (A and AGy) clustered together, whereas the control radiated group (CGy) clustered with both Noggin-treated groups (Figure 7, right panel). These data indicated that Noggin treatment does not drastically affect the gene expression profile of irradiated NSps, whereas ascorbic acid treatment returns the gene expression profile closer to that of non-irradiated controls.

GO pathway analysis was performed with GO-Elite and the biological process pathways overlayed with the heatmap. Genes involved in cell differentiation pathways such as glial cell differentiation, regulation of neuron differentiation, and neuron projection were downregulated in the CGy and Noggin-treated groups when compared to the C and ascorbic-acid-treated samples. These data suggested that treatment of NSps with ascorbic acid reduces radiation-induced astrocyte differentiation.

The number of genes ≥ 1.5-fold changed, with an adjusted *p* ≤ 0.05 for each comparison, can be found in Table 1. Interestingly, there were few genes differentially regulated between the C and AGy conditions as compared to the CGy and NGy groups. Analyzing the overlap in genes differentially expressed between treatments, we observed that only eight genes were shared between the three treatment comparisons: CGy vs. C; AGy vs. CGy; and NGy vs. CGy (Figure 8B). These eight genes (Ccn2, Gfap, Gja1, C1ql1, Slc4a4, Rbp1, Frzb, and Cytl1) are largely involved in astrocyte differentiation and three of them (Ccn2, Gja1, and Gfap) are predicted to significantly interact at the protein level, as shown by the PPI network (Figure 8C). Slc4a4 is expressed predominantly in astrocytes and regulates blood–brain barrier integrity. Gfap is an intermediate filament III protein expressed in astrocytes and is a marker of neuronal differentiation. Gja1 is a connexin expressed in astrocytes [19] and Ccn2 is a growth factor that induces astrogenesis [20]. Slc4a4, Gja1, Ccn2, Frzb, and Gfap were upregulated while Rbp1, C1ql1, and Cytl1 were downregulated in response to the radiation treatment (CGy vs. C) (Figure 3B). Noggin treatment upregulated Ccn2 expression in both N and NGy samples (Figure 8A). In irradiated samples, Noggin treatment reduced the expression of Gfap, Gja1, and Slc4a4 but upregulated Frzb. Importantly, ascorbic acid had no effect on astrogenesis gene expression in non-irradiated NSps (A vs. C) but it did reduce the expression of Gfap, Gja1, Slc4a4, and Ccn2 in the irradiated samples (AGy vs. CGy). This was four out of the five astrogenesis genes upregulated by radiation alone. Altogether, pretreatment of NSps with ascorbic acid is radioprotective and reduces astrogenesis gene expression, which may preserve NPC function and prevent radiation-induced astrogenesis.

## 4. Discussion

Studies have shown that children exposed to whole-brain radiation exhibit a progressive decline in neurocognitive function and predicted intelligence quotient (IQ) [21]. These changes in the development of children demand innovative approaches for reducing neurocognitive toxicity for children requiring whole-brain irradiation [22]. Furthermore, proton radiation demonstrated toxicity reduction by minimizing the exposure of normal brain tissue but only treating focal targets gives similar toxicity to X-ray therapy in whole-brain irradiation [23]. In another laboratory, Ultra-High Dose Rate (UHDR or Flash) irradiation using different radiation sources provides decreased CNS toxicity but more research is needed [24,25,26,27].

Together, we aimed to elucidate a mechanism behind the astrogenesis that occurs in NPCs exposed to radiation in a 3D culture with the goal of designing a high-throughput pre-clinical model to test radiation toxicity in NPCs. This will be used to compare radiation modality effects on NPCs and to pre-clinically screen the safety/benefits of new candidate drugs in combination with radiotherapy or as a radiation CNS toxicity counter measure.

Radiation has been shown to have a significant impact on the behavior of neural progenitor cells (NPCs), which are cells capable of differentiating into neurons or other types of cells in the nervous system. The neurosphere culture system for NPC expansion and self-renewal is an established model proposed by Reynolds BA in 1992 [28]. This classic free-floating culture of NPC aggregates is considered a better system than 2D systems but limitations still exist when assessing effects on NPCs directly. For this reason, our newly designed single-neurosphere culture model shows improvements, not only as a screening tool for potential drugs but also in improving the limitations of the 3D culture standardizing the size of the neurospheres.

Our data suggests that astrogenesis occurs early after radiation in a dose-dependent manner that correlates with the radiation dose when compared to control groups, as previously demonstrated in 2D-cultured NPCs [12]. Furthermore, we implicate BMP signaling in NPC cell fate decisions after radiation and the role in promoting astrogliosis observed after radiation exposure. NPC differentiation is dependent on bone morphogenetic proteins (BMPs) [29,30]. These morphogens are key regulators of NPC differentiation potential toward neuron and glia cell lineages [31,32] and are upregulated in NPCs following insult or disease [33]. In addition, the BMP signaling pathway has been proven to regulate astrocyte commitment in the CNS [34,35]. Up-regulation of proteins involved in this pathway has been extensively shown to direct NPCs to an astroglial fate [36,37,38,39]. Our RNA-seq results further support this, as Notch, BMP2/4, S100b (an early astrocyte marker), and GFAP gene expressions were upregulated at the time of astrogenesis after radiation. Therefore, Tgfb1 in combination with the upregulation of Notch/BMP signaling may be a plausible mechanism for early NPC differentiation into astrocytes as a response mechanism after injury induced by radiation exposure.

Studies have demonstrated that exposure to low-to-moderate doses of ionizing radiation can stimulate the division of NPCs, leading to an increase in the number of cells that can potentially differentiate into neurons and other cells of the nervous system. This has been observed in vitro and in vivo [40,41]. In vitro studies using neurospheres have shown an increase in neurosphere formation and size after exposure to low doses of ionizing radiation [41].

However, exposure to high doses of ionizing radiation has been shown to induce DNA damage and cell death in NPCs, leading to a reduction in their number and potentially affecting their ability to differentiate and participate in tissue repair processes [40,42]. In addition, radiation exposure has been shown to alter the normal pattern of gene expression in NPCs, which can have long-term impacts on the fate and function of these cells [41].

It is important to consider that the effects of radiation on NPCs can be influenced by several factors, including the type and dose of radiation, the stage of development of the cells, and the presence of other environmental stressors [40]. Further research is needed to better understand the complex interplay between these factors and the effects of radiation on neural progenitor cells.

The differentiation of NPCs into astrocytes is a crucial aspect of neural development and is regulated by several signaling pathways, including the Bone Morphogenetic Protein (BMP) signaling pathway [34,43]. Astrocytes play a vital role in maintaining the structural and functional integrity of the nervous system but overactivation of astrocytes can contribute to neuroinflammation and cellular damage, which can lead to the development of various neurological disorders [44].

Modulating the BMP signaling pathway using antioxidants or reactive oxygen species (ROS) scavengers has been proposed as a potential neuroprotective strategy. Antioxidants and ROS scavengers have been shown to regulate the BMP signaling pathway and inhibit the differentiation of neural progenitor cells into astrocytes, thereby reducing oxidative stress and promoting neuroprotection as we demonstrated in the study treating radiation toxicity of NSps with ascorbic acid [45].

Studies have demonstrated that treatment with the antioxidant N-acetylcysteine (NAC) can reduce oxidative stress and decrease the differentiation of neural progenitor cells into astrocytes, leading to improved neuroprotection in models of neurodegenerative diseases [46]. Similarly, treatment with the ROS scavenger Tempol has been shown to inhibit the activation of astrocytes and promote neuroprotection in models of brain injury [47]. Research has revealed varying impacts of ascorbic acid on radiosensitivity in prostate cancer cells and normal prostate epithelial cells [48]. These findings provide evidence for the potential of redox modulators to enhance the effectiveness of radiotherapy while safeguarding normal tissue from injury in brain cancer treatment scenarios.

Despite these promising findings, it is important to note that the precise mechanisms by which antioxidants and ROS scavengers modulate the BMP signaling pathway and impact the differentiation of neural progenitor cells into astrocytes are still not fully understood. Further research is required to determine the safety and efficacy of this approach in humans, as well as to gain a deeper understanding of the complex interplay between BMP signaling, antioxidants, and ROS scavengers in the regulation of neural development and the promotion of neuroprotection. Furthermore, glioma cells cultivated in this system as single neurospheres may be a promising and relevant model for studying tumor mechanistic and signaling pathways, the response to radiotherapy anticancer drugs, and future studies combining gene sets and pathways analysis involved in the formation of glioma [49]; in addition, this study on healthy neurospheres combined with radiotherapy could benefit the creation of new therapies.

## 5. Conclusions

In conclusion, our study demonstrates the potential of single-NSp cultures to recapitulate radiation-induced NPC toxicity and astroglial differentiation. The single-sphere culture methodology could represent an avenue for a high-throughput pre-clinical in vitro model to test radiation toxicity in NPCs, which can be used to compare radiation modality effects on NPCs and pre-clinically screen the safety/benefits of new candidate drugs in combination with radiotherapy.

## Figures and Tables

**Figure 1 cancers-15-04916-f001:**
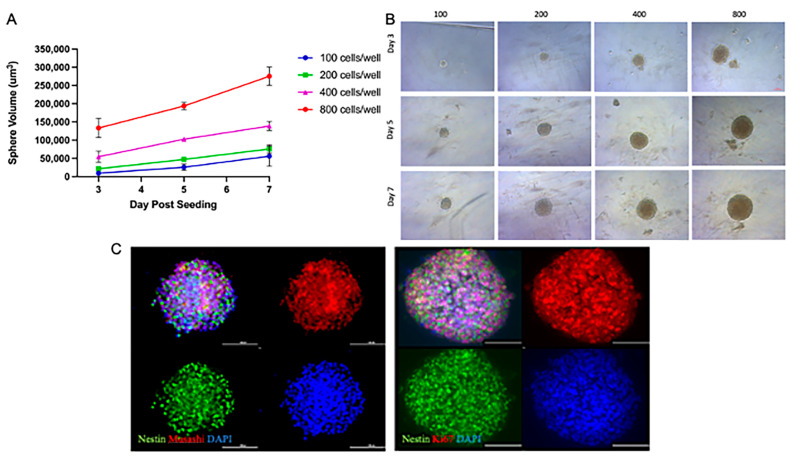
NSp cultures recapitulate NPC-defining characteristics. (**A**) Volume of single neurospheres over time post-treatment with increasing concentrations of cells/well up to 7 days post-seeding (n = 10–15). (**B**) Brightfield microscopy images of neurospheres over time post-seeding with increasing cell concentrations. Representative images are shown (n = 10–15). Scale bar = 100 µm. (**C**) IF images showing staining for nuclei with DAPI (blue), stemness with Nestin (green), and Musashi (left; red), cellular proliferation with Ki67 (right; red). Representative images are shown (n = 5–6). Scale bar = 100 µm.

**Figure 2 cancers-15-04916-f002:**
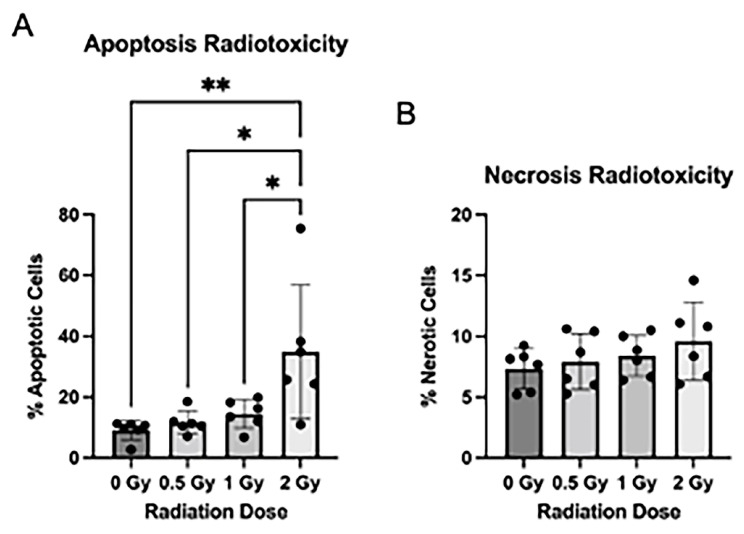
Radiation induces apoptosis after radiation exposure: (**A**) Neurospheres were dissociated and stained with Annexin V-APC + PI for flow cytometry. The number of apoptotic cells post-radiation treatment was quantified by measuring the proportion of Annexin V+ and PI+/PI− cells (n = 5–6); (**B**) The number of necrotic cells post-radiation treatment was quantified by measuring the proportion of Annexin V− and PI+ cells (n = 5–6 independent replicates per condition) (mean ± SD, * *p* < 0.05, ** *p* < 0.005).

**Figure 3 cancers-15-04916-f003:**
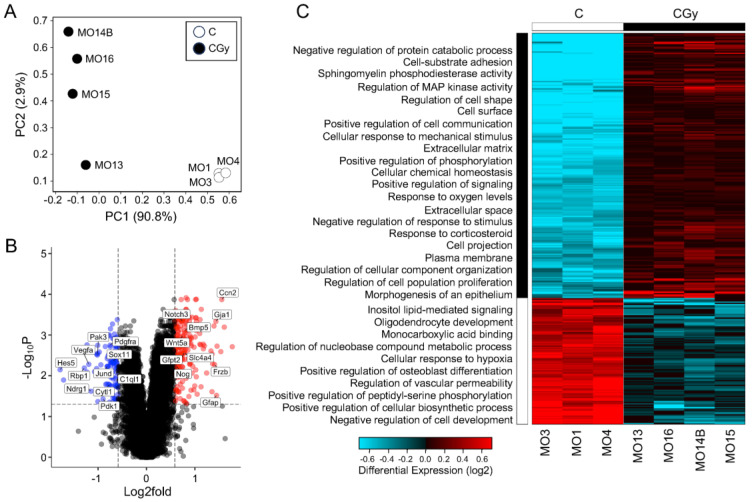
The transcriptome of NSps changes drastically after radiation and indicates an increase in NPC differentiation: (**A**) Principal component analysis (PCA) of bulk RNA-seq from neurospheres treated as control (C) (n = 3) or with 1 Gy radiation (CGy) (n = 4) samples; (**B**) Volcano plot of all genes 1.5-fold upregulated (red dots) or downregulated (blue dots) meeting an adjusted *p*-value threshold < 0.05. A subset of genes are labeled; (**C**) Hierarchical clustering heatmap of differentially expressed genes indicating enriched GO: biological processes in neurospheres exposed to radiation as compared to controls.

**Figure 4 cancers-15-04916-f004:**
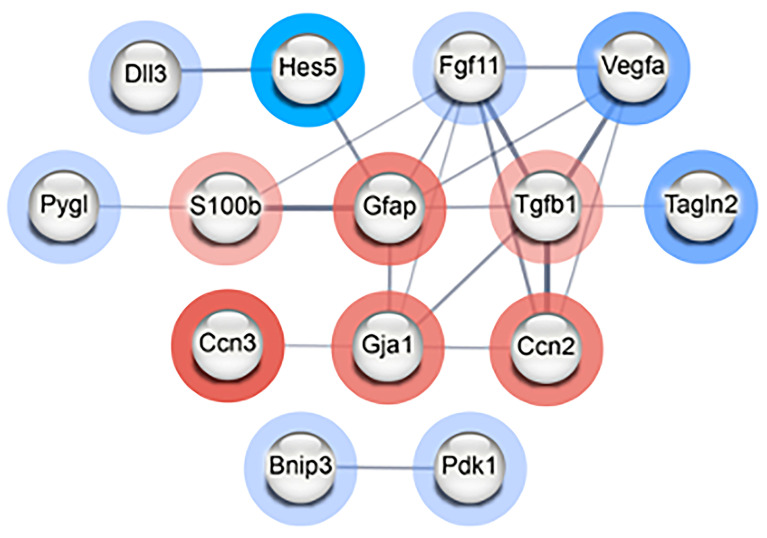
Radiation modulates genes involved in differentiation of NPCs in neurospheres. STRING PPI network of interconnected nodes with an interaction score of at least 0.4 created from the list of top 20 most upregulated and downregulated genes in radiation-treated (CGy) vs. control (C) neurospheres. Genes/proteins outlined in red were upregulated and those outlined in blue were downregulated. The intensity of the colored ring indicates the fold change. The edge line thickness indicates the strength of data supporting the interaction. The PPI enrichment *p*-value = 4.24 × 10^−5^ indicates that there are more predicted PPI interactions amongst this set of genes/proteins than would be expected randomly.

**Figure 5 cancers-15-04916-f005:**
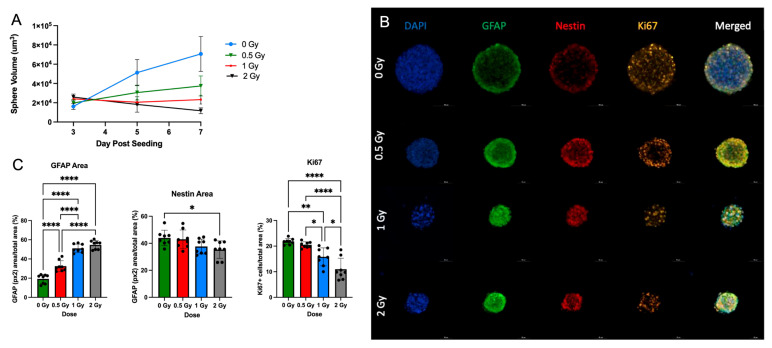
Radiation promotes NSC differentiation in single NSps: (**A**) Volume of single neurospheres over time post-treatment with or without (blue) increasing doses of radiation; 0.5 Gy (green), 1 Gy (red), and 2 Gy (black) (n = 10–15); (**B**) Progressive generation of astrocytes in co-staining of GFAP (green), Nestin (red), Ki67 (orange), and DAPI (blue) single neurospheres; (**C**) GFAP+, Nestin+ area/total stained area and Ki67 cells/total stained area in all groups (mean ± SD, * *p* < 0.05, ** *p* < 0.005, **** *p* < 0.0001) (n = 8–10).

**Figure 6 cancers-15-04916-f006:**
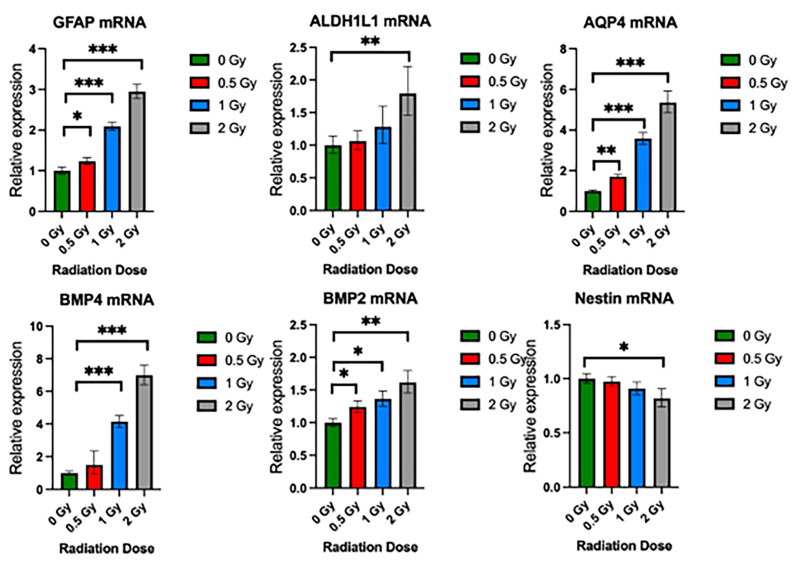
Radiation promotes upregulation of NSC differentiation genes. Relative gene expression of GFAP, ALDHL1, AQP4, BMP4, BMP2, and Nestin in neurospheres exposed to 0 Gy, 0.5 Gy, 1 Gy, and 2 Gy radiation. Mean ± SE of relative expression (2^−ΔΔCt^) for radiation-treated and control (mean ± SD, * *p* < 0.05, ** *p* < 0.005, *** *p* < 0.001 (n = 8–10).

**Figure 7 cancers-15-04916-f007:**
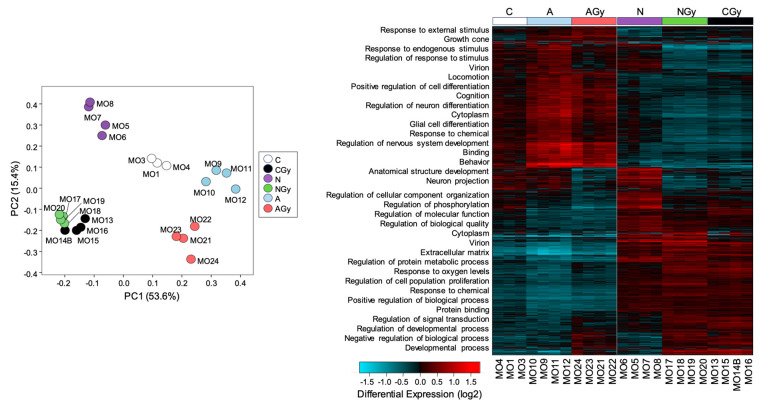
Single-neurospheres culture for radiation toxicity screening. Principal component analysis of libraries sequenced for RNA-seq for neurospheres and Hierarchical clustering analysis of differentially expressed genes from control (C), radiation-treated neurospheres (CGy), ascorbic-acid-treated neuropsheres (A) ascorbic-acid- and radiation-treated neurospheres (AGy), Noggin-treated neurospheres (N) and Noggin- and radiation-treated neurospheres (NGy).

**Figure 8 cancers-15-04916-f008:**
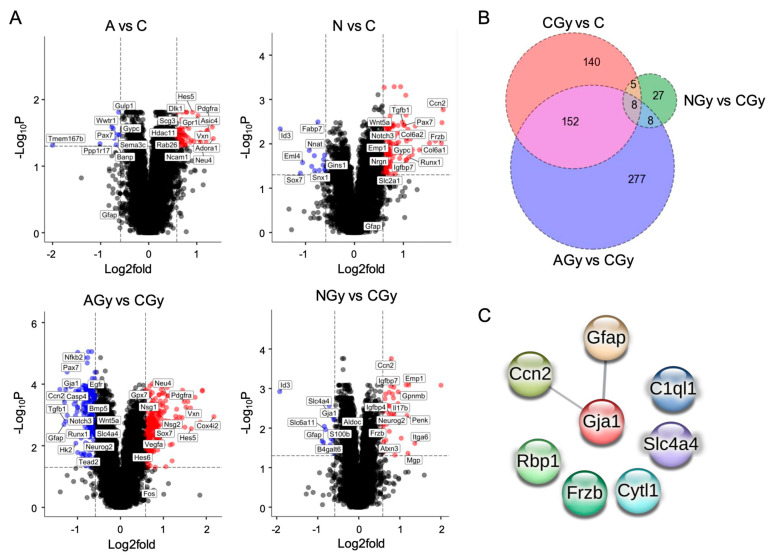
(**A**) Volcano plots of RNA-seq data from neurospheres of the indicated treatment group comparisons. All genes ≥1.5-fold upregulated (red dots) or downregulated (blue dots) meeting an adjp-value threshold ≤ 0.05 are indicated. A subset of genes is labeled. (**B**) Venn diagram shows the number of significant genes (adjp-value ≤ 0.05) with a fold change ≥ 1.5 in CGy vs. C, AGy vs. CGy, and NGy vs. CGy comparisons. (**C**) STRING predicted protein–protein interaction (PPI) network based on the 8 dysregulated genes shared between the CGy vs. C, AGy vs. CGy, and NGy vs. CGy comparisons with an interaction score of at least 0.4. The edge line thickness indicates the strength of data supporting the interaction. The PPI enrichment *p*-value of 0.0489 is significant, which indicates that the input proteins have more predicted interactions than would be expected randomly.

**Table 1 cancers-15-04916-t001:** Number of differentially expressed genes identified as ≥1.5-fold change and adjp ≤ 0.05.

	C		CGy		N		NGy		A		Agy	
	Up	Down	Up	Down	Up	Down	Up	Down	Up	Down	Up	Down
C												
CGy	209	97										
N	138	15	132	53			150	52	322	247	329	209
NGy	244	147	35	13								
A	56	10	403	371			522	401			13	10
AGy	65	33	224	222			282	241				

## Data Availability

Data supporting the findings of this study are available within the article and its Appendix A.

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
