# Peer review of "Development of a Single-Neurosphere Culture to Assess Radiation Toxicity and Pre-Clinical Cancer Combination Therapy Safety"

_cancers, 2023, doi:10.3390/cancers15204916_

Round 1

Reviewer 1 Report

The present study was to establish a neurosphere culture platform to screen the idealist or candidate drugs. The concept of the study is innovative and urgent for the radiomedicine field. The following are some issues that need to be revised. 

1. The abstract is too extended and far exceeds 200 words. Some paragraphs may be combined or replaced into the introduction, M&M, or result section. 

2. Some of the descriptions in the abstract are the same as in the introduction section. This issue must be revised.

3. The information on the materials was not shown. 

4. The immunostaining photos of Figure 1 are hard to review due to the resolution. Some neuronal markers did not explain the meaning in the beginning, e.g. DCX. 

5. It is unclear whether the culture experiment (neurosphere-related) was done once. And only n=10-15 in Figure 1 and n=5-6 in Figure 2. I think this is not very powerful for the study.

6. The results demonstration of Figure 3 is not completed. 

7. Figure 4 lists the top 20 genes, but there are only 14 genes in the result.

8. The authors did not describe and explain what MO means in the experiments but only addressed that MO2 was the outlier cell.

9. It is unclear about ascorbic acid and noggin dosages on Neurospheres.

10. Some typos. For example, in the discussion section, line 497.

I think that there is no major English language issue.

Author Response

Reviewer #1

The present study was to establish a neurosphere culture platform to screen the idealist or candidate drugs. The concept of the study is innovative and urgent for the radiomedicine field. The following are some issues that need to be revised. 

1. The abstract is too extended and exceeds 200 words. Some paragraphs may be combined or replaced into the introduction, M&M, or result section. 
Response: We have streamlined the abstract so that it is now within the 200-word limit (195 words).

2. Some of the descriptions in the abstract are the same as in the introduction section. This issue must be revised.

Response: We appreciate the reviewer’s comments and have revised the abstract and introduction to address this concern.

3. The information on the materials was not shown. 

Response: Thanks so much for the comments. Now the methods section was revised

4. The immunostaining photos of Figure 1 are hard to review due to the resolution. Some neuronal markers did not explain the meaning in the beginning, e.g. DCX. 

Response: Thanks so much for detecting the issue and a new picture with better quality is included and also the markers not related to this manuscript were deleted for better understanding for the reader.

5. It is unclear whether the culture experiment (neurosphere-related) was done once. And only n=10-15 in Figure 1 and n=5-6 in Figure 2. I think this is not very powerful for the study.

Response: Thanks so much for the comment and we realized that was not well explained in the methods section. The new paragraph includes more detail about the n number.

6. The results demonstration of Figure 3 is not completed. 

Response: We appreciate the reviewer’s comment and to address this have added more clarifying language to the Figure 3 result section.

7. Figure 4 lists the top 20 genes, but there are only 14 genes in the result.

Response: We apologize to the reviewer for the confusion. The top 20 upregulated and top 20 downregulated genes were used as input into STRING, however only 14 genes were connected in a network, which are the 14 genes shown and discussed. We have clarified this in the text.

8. The authors did not describe and explain what MO means in the experiments but only addressed that MO2 was the outlier cell.

Response: We apologize that this was inadequately explained. MO# is simply the sample identifier/name used during this experiment. We have clarified this in Materials & Methods section 2.8.

9. It is unclear about ascorbic acid and noggin dosages on Neurospheres.

Response: We appreciate the reviewer noticing this crucial detail and have added this information to section 3.5.

10. Some typos. For example, in the discussion section, line 497.

Response: We thank the reviewer for the comment and have further edited for proper English language, grammar, punctuation, and spelling to address this concern.

Reviewer 2 Report

In this article the authors have assessed the radiotoxicity in spheroids grown from neural progenitor cells, and analyzed biological markers relevant to growth, apoptosis, necrosis and survival through immunoassays, RT-qPCR and transcriptome techniques. The authors also conducted gene-protein informatics and predict relations and evaluated role of antioxidants in reducing radiotoxicity. Similar studies on spheroids and organoids have been performed in previous studies and studies exploring role of antioxidants in radiotoxicity have been explored (doi.org/10.3390/cancers12020469, doi.org/10.3390/cells11193106, doi.org/10.1158/0008-5472.CAN-16-0785). The motivation is clear, and the authors have performed appropriate analysis to explain the mechanism and conclusions of their study. Although the data presented in this article is within the scope of the journal, the authors can bolster the results by including more in-vitro data (non-spheroid) supporting their hypothesis. The authors can discuss if they have observed similar biomolecular changes in a xenograft model compared to neurospheres.

Minor editing required.

Author Response

Reviewer #2

In this article the authors have assessed the radiotoxicity in spheroids grown from neural progenitor cells, and analyzed biological markers relevant to growth, apoptosis, necrosis and survival through immunoassays, RT-qPCR and transcriptome techniques. The authors also conducted gene-protein informatics and predict relations and evaluated role of antioxidants in reducing radiotoxicity. Similar studies on spheroids and organoids have been performed in previous studies and studies exploring role of antioxidants in radiotoxicity have been explored (doi.org/10.3390/cancers12020469, doi.org/10.3390/cells11193106, doi.org/10.1158/0008-5472.CAN-16-0785). The motivation is clear, and the authors have performed appropriate analysis to explain the mechanism and conclusions of their study. Although the data presented in this article is within the scope of the journal, the authors can bolster the results by including more in-vitro data (non-spheroid) supporting their hypothesis. The authors can discuss if they have observed similar biomolecular changes in a xenograft model compared to neurospheres.

 Response: We thank the reviewer for the comment and suggestion. The present study introduces single sphere culture methodology that could represent an avenue for a high-throughput pre-clinical in vitro model to test radiation toxicity. Other studies explored the mechanisms in other cell types and cancer cells which really support the same idea for this study. We really appreciate the recommendations from the reviewer, and we added some more clarification and support during the discussion. Also have further edited for proper English language, grammar, punctuation, and spelling to address this concern.